# Evaluation of Health Information System (HIS) in The Surveillance of Dengue in Indonesia: Lessons from Case in Bandung, West Java

**DOI:** 10.3390/ijerph17051795

**Published:** 2020-03-10

**Authors:** Lia Faridah, Fedri Ruluwedrata Rinawan, Nisa Fauziah, Wulan Mayasari, Angga Dwiartama, Kozo Watanabe

**Affiliations:** 1Parasitology Division, Department of Biomedical Science, Faculty of Medicine, Universitas Padjadjaran, Bandung 45363, Indonesia; nisa@unpad.ac.id; 2Foreign Visiting Researcher at Department of Civil and Environmental Engineering, Ehime University, Matsuyama, Ehime 790-8577, Japan; 3Department of Public Health, Faculty of Medicine, Universitas Padjadjaran, Bandung 45363, Indonesia; f.rinawan@unpad.ac.id; 4Anatomy Division, Department of Biomedical Science, Faculty of Medicine, Universitas Padjadjaran, Bandung 45363, Indonesia; wulan.mayasari@unpad.ac.id; 5School of Life Sciences and Technology, Institut Teknologi Bandung, Bandung 40132, Indonesia; dwiartama@sith.itb.ac.id; 6Department of Civil and Environmental Engineering, Ehime University, Matsuyama, Ehime 790-8577, Japan; watanabe_kozo@cee.ehime-u.ac.jp

**Keywords:** dengue surveillance, health information system, Bandung

## Abstract

This study was performed to evaluate the health information system regarding the dengue surveillance system in Indonesia. Major obstacles to the implementation of an effective health information system regarding dengue cases in Bandung are examined, and practical suggestions on measures to overcome them are discussed. The study utilized a mixed-method research design using qualitative approaches: document analysis, key informants and focus group interviews. Thirty key informants were selected, comprised of policymakers, senior managers, and staff at the Ministry of Health. Data from documents and transcripts were evaluated through a modified Institutional Analysis and Development (IAD) framework described by Ostrom. Through this study, we have identified several issues that hinder the effective implementation of the health information system in the case of dengue in Bandung. In the end, we propose several recommendations for reform that encompasses motivational, strategic, and structural approaches to each component of the analysis. Through evaluation of the health information system for dengue surveillance in Indonesia, we conclude that well-coordination in multi-level governance in a country as large as Indonesia is the key in the implementation of the health information system in different levels of agencies. Furthermore, the adaptability of human resources in adopting a new information system also plays an important part.

## 1. Introduction

Dengue fever is an infectious disease caused by flavivirus which is carried by Aedes mosquito vectors (*Aedes aegypti* and *Aedes albopictus*). It is an important mosquito-borne disease in the world and has been categorized as a “neglected tropical disease” [1]. Nearly 2.5 billion people live in dengue-endemic regions globally, and each year an estimated 50–100 million people are infected, resulting in 500,000 hospitalizations and approximately 20,000 deaths [1]. Of the global population at risk, more than 70%—or about 1.8 billion people—live in the Asia-Pacific region and as such, Asians contribute the most to the overall burden of dengue [2].

Indonesia is no exception when it comes to the threat of dengue. As a country with a tropical climate and high humidity, Indonesia is favorable for transmission of vector-borne disease [3]. Since the first case of dengue reported in Jakarta and Surabaya in 1968, the disease has been expanding in incidence and geography [4]. The annual incidence of dengue fever has grown from 0.05/100,000 in 1968 to ~35–40/100,000 in 2013, with the highest epidemic occurred in 2010 (85.7/100,000) [3]. The spread of dengue also expands in terms of geography, and it is likely to be hyperendemic (i.e., multiple co-circulating serotypes) nationwide in Indonesia [4]. Therefore, it is no surprise that the cost associated with dengue in Indonesia is considerably burdening. In a study by Shepard and colleagues, it is estimated that annual cost affiliated with dengue infection in Indonesia in 2001–2010 reached US$ 323 million [2].

To counter the burden of dengue infection, the Indonesian government has built dengue programs vertically from sub-district, district, provincial, to national levels and incorporates vector control, public health campaigns and education, training and research, and also most importantly, epidemiological surveillance [5]. Included in this effort of dengue surveillance is the establishment of the Health Information System (HIS), which collects clinical and epidemiological data from the health sector and other relevant sectors to obtain information for health-related decision-making [6].

However, Indonesia is a country with a vast region—with 257.5 million inhabitants and 17,500 islands spread across the equator—with varied geography, biodiversity, population densities, and characteristics [3]. Moreover, it is a developing country—and it has been known through previous studies that there is still limited evidence on how health information systems have contributed to improved health outcomes, and to advancing the state of the poor in developing countries [7]. Therefore, this calls for the need for evaluating the effectiveness of HIS in Indonesia.

Unfortunately, studies evaluating Indonesian HIS concerning dengue infection are limited. The evaluation of HIS in Indonesia was done only once in 2007 [8] (Kementerian Kesehatan, Indonesia, 2012). Thus, we attempt to evaluate HIS in regards to dengue surveillance policy in Indonesia. In this study, we use a case from Bandung, the capital city of West Java province which is one of the most dengue-prevalent regions in Indonesia [3]. Through this study, we utilize the longitudinal approach in order to provide insight of how HIS in case of dengue infection is implemented in Indonesia and what are the major obstacles and practical suggestions to improve the system.

## 2. Literature Review

### 2.1. Health Information System

Health Information System (HIS) is a system that collects data from the health sector and other relevant sectors in public health, analyses these data and ensures their overall quality, relevance and timeliness. Subsequently the analyzed data are converted into useful information for health-related decision-making. In practice, this data could be used at different levels of the health care system, such as individual, health facility, population and public health surveillance level [6]. Generally, such an information system is required in the planning and implementation of health care interventions to improve health systems and attain better health. It also helps to assess the health needs of populations and evaluate the effectiveness and coverage of health programs [9,10].

The need for an integrated HIS itself grew decades ago in relation to the emerging and increased complexity of health management and public health policy, particularly in developing countries [11]. In these countries which includes Indonesia, the HIS becomes a crucial instrument. It provides a coordinating framework to address various health problems, including rampant diseases [12]. However, the implementation of HISs in developing countries is not without problem. In several studies involving developing countries, there have been several major obstacles discovered. First, it is acknowledged that issues surrounding the implementation of HISs often come not from the technicalities of the system itself but from the social, political, economic, and public health context in which the HIS is meant to operate [13]. Second, the capabilities of human resources (health staff) to implement a new health management information system also determines the success of the program [14]. Third, it is noted that no matter how good the health information system is, without commitments and leaderships of the people managing it, the program will not be able to bring satisfactory results [15].

If not implemented properly, a HIS can negatively affect the provision of healthcare services. Consequently, to ensure a high-quality health system, continuous evaluation and upgrades of existing HISs are as important as the implementation [16].

### 2.2. Evaluation Framework of HIS: Institutional Analysis and Development (IAD) Framework

Evaluation is the act of measuring or exploring properties of a health information system (in planning, development, implementation, or operation), the result of which informs a decision to be made concerning that system in a specific context [17]. There is no ideal and specific way of evaluating healthcare systems. Evaluation methods can be complex, single or combined, and with a lot of variables. Frameworks mainly describe or measure features or categories of HISs that will guide towards the improvement of the system and can be done at different time points during the development, implementation and post-implementation of the HISs and taking into consideration the different stakeholders that will interact with or benefit from the system [16].

One of such frameworks that can be utilized to evaluate HISs is the Institutional Analysis and Development (IAD) framework, which was described by Ostrom (1994). It is a framework that has been used in the health sector through what is called as Health Policy Analysis Framework [18]. In principle, IAD is a systematic method for organizing policy analysis activities that is compatible with a wide variety of more specialized analytic techniques used in the physical and social sciences. It helps analysts comprehend complex social situations and break them down into manageable sets of practical activities [19]. In a way, IAD provides a means for inquiring into a subject by bringing explicit attention to the relevant variables and the questions one may want to ask [20]. When applied rigorously to policy analysis and design, researchers would have a better possibility to avoid the oversights and simplifications that lead to policy failures [19].

To analyze complex situations, IAD breaks down such circumstances into an ‘action arena’—the core of this framework. The action arena (see Figure 1) is composed of an action situation and actors. The action situation refers to a social space where the actors interact, solve the problem of commons, and exchange goods and services; the actors are those who participate in the situation [22]. Action arenas can be everywhere: in the home, environment, market, company, and, in this case, public health institutions.

In IAD, the behavior of participants in the action situation is influenced by three sets of exogenous variables: the rules-in-use, the biophysical and material world, and the community [20]. First, rules-in-use is something that was set by authorities and was agreed upon by the members of the community. Within an institutional framework, an institution is defined as a system of established and embedded social rules that structure social interactions, which can be anything from language, law to traffic conventions and table manners. Institutional rules can either be formal, textual and formed by a process of conscious design (as in Law and regulations), or informal and emerging from social practices and collective actions [23], Second, biophysical and material conditions relate to any non-human components of the system: infrastructure, technology, money, diseases, ecological factors, etc. These factors may influence the way rules are implemented. Third, the community relates to accepted behavioral values and knowledge within participants of the system (in this case, HIS). The variability of community attributes depends on the homogeneity of the community’s preferences, the structure of the community, and the level of inequality among those impacted by the system [24].

The interactions that emerge from the action arena result in outcomes [24]; the IAD framework describes three successive and interrelated levels of outcomes: operational, collective, and constitutional choice. Operational-level outcomes are day-to-day activities that affect the physical world directly. Collective choice-level outcomes are the rules established by decision-makers to shape operational-level activities. Finally, constitutional-level outcomes are the results of decisions about how collective-choice actors are selected and which patterns of interaction will define the relationship among participants of the collective-choice body [25].

These outcomes will then be evaluated through certain criteria. This evaluation will provide feedback to both the action arena and the exogenous variables, to improve the institutional settings so that better outcomes can be produced [24]. In evaluating public policy, there are two approaches to employ IAD: (1) IAD as a diagnostic tool can move backwards along the diagram to revise policy, evaluate results, understand informational structure and policy incentives, as well as construct changes. This is particularly useful in analyzing a well-established and implemented policy; (2) IAD can also move forward to define issues and goals and by using IAD as a form of political-economic activity. This approach is useful in developing new initiatives and comparing different policy alternatives [19].

## 3. Research Methods

### 3.1. Ethical Declaration

This study was approved by the Medical Research Ethics Committee of Faculty of Medicine Universitas Padjajaran, Bandung Indonesia with approval number 071711. The informants were aware of this study and provided informed consents accordingly.

### 3.2. Study Area

While covering a national-based HIS, the study was conducted in the city of Bandung, West Java—one of the regions in Indonesia with a high prevalence of dengue infection [3]. In particular, we chose Bandung city due to its unique characteristics of being relatively isolated and also the only high-altitude region in Indonesia with constant endemicity of dengue each year [26,27] (Komariyah, 2006 and Suganda, 2007). These characteristics led to an ongoing discussion of Bandung’s appointment as a pilot city in dengue intervention projects in Indonesia. The city encompasses 30 districts and 151-subdistricts [28], accommodating a total population of around 2.5 million people [29]. Health services in Bandung are delivered through the public and private health sectors—which includes 33 hospitals, 120 community health centers, and 248 clinics [30].

### 3.3. Data Collection

This study utilizes a mixed-method research design to assess the performance of HIS in Bandung. We employ a qualitative approach, combining document analysis alongside and focus group interviews involving key informants.

In document analysis, data were collected from various published and unpublished documents dated from 2014 onwards. This includes National Development Plans, Indonesian Ministry of Health policies, related reports from Government and other agencies, and research reports.

Meanwhile, we purposively select a total of 30 key informants. These informants comprised of policymakers, senior managers, and staff of the Indonesian Ministry of Health. Participants were recruited through personal contact. All interviews were audiotaped and transcribed.

### 3.4. Data Analysis

The data were evaluated through the Health Policy Analysis Framework, which adopted Ostrom’s IAD framework. In particular, we used a modified IAD framework to analyze and evaluate the implementation of HIS within the current health policy in Indonesia. In so doing, a content analysis was done through a deductive data coding process, which generic themes were derived from the IAD conceptual framework (biophysical and material conditions, rules-in-use, and community attribute). In this paper, the action arenas were analyzed on three different levels: constitutional, collective, and operational levels. During our thematic analysis, however, we also identified several specific issues, namely cross-scale coordination, system interoperability, uniformity of data reporting, and stakeholder engagement, which link with the larger thematic categories and provide hints to develop a better policy recommendations. 

## 4. Results and Discussions

### 4.1. Material Condition: National Health Information System of Indonesia

To monitor the trend of dengue infection and to ensure early vigilance towards dengue outbreak/extraordinary cases, the Indonesian Ministry of Health has launched Guidelines for Dengue Hemorrhagic Fever Prevention and Control in 2017. One of the important measures described in the guideline is dengue case information system (see Figure 2) [31].

In the dengue prevention and control guideline described by the Ministry of Health, there are several types of the report regarding dengue cases [31]:(1)W1 is a report that is created upon encountering dengue outbreak or extraordinary events;(2)W2 is a weekly epidemiological report that is consisted of several priority infectious diseases including dengue;(3)K-DBD is a routine monthly report of dengue case;(4)DP-DBD reports detailed record of individual dengue patients; and(5)KD-RS is reports made by hospitals and other health facilities such as clinic and private practitioners upon diagnosis of dengue among their patients

In Indonesian dengue information system (see Figure 2), cases of dengue enter the information system through two methods: either through reports made by hospitals and health facilities (KD-RS) or through reports made by community members to Community Health Center (Puskesmas) (W1 and W2-DBD, DP-DBD, and K-DBD). Either way, these two institutions are connected as hospitals and health facilities have to send copies of their reports to the Community Health Center [31]. These reports can be inputted in both computerized and manual format, depending on available infrastructure in these facilities.

Reports from hospitals, health facilities, and the Community Health Center are then processed in the Regional Public Health Office—starting from the level of city/district and subsequently in the level of province. In the Regional Public Health Office, data regarding dengue cases are kept by a work unit that specializes in keeping data and information, the Data and Information Center (Pusdatin). Surveillance unit uses the data from the Data and Information Center and forwards it to the implementing unit which operates dengue intervention programs [31]. Finally, all dengue cases that have been recapitulated by the Public Health Office of each province are reported every month to the Ministry of Health Indonesia. Regarding dengue, there are two relevant directorate generals which will use this information: Directorate General of Disease Control and Directorate General of Environmental Health [31].

In general, the Indonesian dengue information system that is described by the Ministry of Health assures that information of dengue case reports flows unidirectionally. In this system, dengue information is summarized as it moves upward the institutional ladder. However, our research has found a stark contrast in the flow of dengue case information in the existing situation of the study in Bandung (see Figure 3).

One difference discovered upon interview and group discussion is the lack of coordination among hospitals and health facilities with the Community Health Center. Hospitals and health facilities rarely send copies of their report (KD-RS) directly to the Community Health Center—instead they delegate this responsibility to each patient, who has to individually report to the nearest Community Health Center. This has caused many cases of underreporting, as individual patients often forget to report to the Community Health Center. Moreover, it also causes a disparity in data held by the Community Health Center and the city/district Regional Health Office.

A discrepancy also happens across the levels of the Regional Public Health Office. The Data and Information Center, which is supposed to collect both computerized and manual forms of dengue records only gather the computerized data due to technical difficulties in recapitulating manual reports. Because of this data incompleteness, dengue implementation units take the initiative to summarize the manually-inputted dengue reports by themselves—causing lack of correspondence in the number of dengue cases recorded by different units within the Regional Public Health Office (see Figure 4).

The most significant discrepancies are perhaps found in the relationship between the Community Health Center, Ministry of Health, and surveillance unit of Regional Public Health Office. While in the 2017 Dengue Information System there is no direct relationship between these three entities, the existing condition shows that there is a new information route between the three of them through the direct computerized weekly W2 report, as an implication of Early Warning Alert and Response System (EWARS). EWARS is a web-based early warning system established by the Ministry of Health in 2009, aiming to detect health events of significance for public health and health security. In this system, the Community Health Center directly reports to a national level server and this data is forwarded to the surveillance unit in corresponding regions which would perform the appropriate verification and response as required [32]. Therefore, the data utilized by the surveillance unit is different from that of the implementing unit and Data and Information Center, further exacerbating the lack of data integration within the Regional Public Health Office (see Figure 4).

Our study also found that the complexity of the dengue information system in Indonesia is also attributed to differences in the nomenclature used by Data and Information Center, surveillance unit, and implementing unit to categorize dengue cases. The Data and Information Center simply uses the terms ‘new cases’ and ‘old cases’ of dengue. New cases are data collected from hospitals, while old cases are data collected from the Community Health Center. Meanwhile, the surveillance unit refers to all cases of dengue whether confirmed or yet-to-be as dengue fever suspect. On the other hand, the implementation unit uses international standard ICD10. In ICD10, cases of dengue are grouped according to their respective medical symptoms into four categories: Dengue Fever, Dengue Hemorrhagic Fever, Dengue Shock Syndrome, and Expanded Dengue Syndrome [31]. The distinction of nomenclature used in information systems of different units causes the lack of interoperability of data amongst these units, with the result that data collected by the Data and Information Center has not been used optimally by other institutions in the health system.

### 4.2. Rules-In-Use

The fundamental basis of the health information system in Indonesia lies in article 167 and 168 of the Indonesian Health Act no 36, 2009 regarding health management and health information. Referring to these articles, health information is a part of an integrated health management system that aims to ensure the highest degree of health for everyone. Specifically, health information is required to enable effective and efficient health efforts. This health information is then organized through a health information system that covers various sectors [33]. To carry out the Indonesian Health Act, the president of Indonesia subsequently delegates the responsibility of arranging required organization and work procedures to the Ministry of Health through the Presidential Decree no 47, 2009 [34].

With the said authority, the Indonesian Ministry of Health has established the Data and Information Center as declared in Chapter XII of the Indonesian Republic Minister of Health Decree no 1144/MENKES/PER/VIII/2010. The Data and Information Center is assigned to manage health statistics, to analyze and to disseminate information, and to develop an information system and data bank [35]. Responsibilities of the Data and Information Center is further expounded in the Ministry of Health Regulation no 64, 2015, which explicitly states that entailed among these responsibilities is to draft technical and implementation policies [36], which includes the information system standards, regulations, and guidelines [37].

Despite the clear fundamentals of rules regarding the health information system provision in Indonesia, we found that the implementation is severely lacking. First, there are no clearly set guidelines regarding the health information system including in the case of dengue. The lack of clearly defined guidelines has caused differences in the data collected across different stakeholders, even across different units in the same stakeholders such as in the case of the Regional Public Health Office. Moreover, there is little to none of the references that can be utilized by developers of health information system application regarding the standards of collected data [38]. This causes the lack of synchronization of the data collected by various stakeholders which sometimes employ health information system applications built by different developers.

### 4.3. Stakeholders and Action Situation

In this study, the constitutional level encompasses the government and the Ministry of Health. As stated in the Indonesian Republic Minister of Health Decree no 192/MENKES/SK/VI/2012 and subsequently in the Ministry of Health regulation no 97/2015, the Indonesian government standardizes, manages, and develops the nationwide health information system while also facilitates the development of the regional health information system. As a consequence, the provincial governments manage and develop the health information system on the province scale while the district/city governments manage and develop the health information system on the city/district scale [39]. Unfortunately, this statement gives rise to different perceptions in each level of government. On the level of the national government, this regulation aims to accommodate regional autonomy while also keeping the health information system integrated nationally. However, the statement is perceived differently on the regional level—it is misunderstood as unlimited authority to construct its own health information system, causing variation in the form and the implementation of the health information system across regions. This situation leads to the lack of uniformity of reporting data and ultimately hinders the optimal utilization of the data [40].

The collective level is responsible for designing internal procedures and assisting the implementation of the health information system. Therefore, good communication of decisions and rules is crucial in determining the success of the health information system, together with other characteristics of the organizations such as the institutional size, the heterogeneity of the participants, the perceived benefits received, the outcomes expected, and the monitoring techniques applied [41]. In this level, the Data and Information Center of the Ministry of Health is responsible for monitoring and evaluating the health intervention programs, including regarding dengue. However, we found that the health information system at local hospitals in the district/city level is regulated under a different authority—the Data and Information Center of the local government. Thus, the Ministry of Health could not interfere with the system at these hospitals. This conflict of authority lasted until 2018; in 2019, local hospitals are placed structurally under the authority of the Ministry of Health.

The operational level corresponds to daily activities in the health information system. There are several factors which influence the implementation of the health information system, such as the understanding of health officers regarding the system, motivation, workload, and cooperation [42]. In this study, the operational level focuses on health staff at the hospitals and the Community Health Center. Through interview and group discussion, we identified that the main issue on this level is the abundance of reports that have to be done for each case of dengue. As seen in the existing condition of the health information system of dengue (see Figure 3), a health facility has to file several kinds of reports for each case of dengue. For example, the staff at the Community Health Center has to file five different reports on dengue: W1, W2, DP-DBD, K-DBD, and direct computerized W2 report—with each of them having their own standards and specifications. Moreover, reports to different units often employ different nomenclature to categorize cases of dengue. To exemplify, reports to the implementing unit utilized ICD10 nomenclature, while the direct computerized W2 reports which will be forwarded to surveillance unit categorizes all cases as dengue suspect. Together, these have caused confusions which overwhelm the health staff.

### 4.4. Recommendations for Reform

Overall, the situation regarding the information system in the case of dengue in Indonesia can be summarized as follows (see Figure 5):

The existing condition has resulted in several issues that hinder the implementation of the health information system in the case of dengue in Indonesia and deviate the course from its original aims. Some of these issues include:(1)Insufficient government evaluation of the activities of health institutions and their compliance with regulations;(2)Unequal implementation of the health information system on the national and regional levels;(3)Lack of internal and external measures to monitor compliance with rules and a lack of consequences of defying them;(4)Low motivation and involvement of person-in-charge at the operational level.

Reflecting upon these findings as well as comparing to the health policy analysis framework in a similar case [18], we propose several recommendations in the case of the dengue health information system in Indonesia. These recommendations involve three approaches that can be implemented to solve public dilemmas, which are motivational, strategic, and structural approaches [18].

In terms of material conditions, the health information system requires more interoperability across agency levels and units. This can be done by having a mutually-agreed nomenclature to organize the cases of dengue. Moreover, access towards computerized technology whether in terms of operation, data processing, and the data itself need to be more equitable among different levels of users. The health information system could also be greatly improved by feedback on the applicability and utilization of the data—therefore, assessment of the system needs to be spelt clearly to all stakeholders.

In terms of rules-in-use, informal and local regulations need to align with the formal government regulations. Each of the regional stakeholders (the Regional Public Health Office, hospitals, and the Community Health Center) needs to agree upon the same informal/local regulation so that its implementation will work per the guidelines set at the national level. This also has to be complemented by a clear monitoring and evaluation scheme, as well as an incentive/disincentive mechanism to assure compliance. The incentives do not only relate to financial incentive, but also informational incentive whereby those who comply with the rules-in-use will have better access to data and information produced by the health information system.

Both the material and institutional approaches are needed to address the cross-scale coordination and management issues. If the national government can develop a digital health information system platform that is interoperable with the systems developed at the provincial and district levels, while at the same time implement a strong incentive for the use of this platform across the actors, then the government will be able to build a conducive health information ecosystem needed.

In terms of community attribute and action arena, all stakeholders need to be involved in a more equitable manner and through a participatory approach. The capacity of staff members to operate and to use the health information system is crucial in the case of dengue. Therefore, organizational structures will need to adjust in a way that encourages the implementation of the national health information system, in such a manner that will push them to gain better access and wider incentives out of the health information system. Moreover, an independent working body to ensure the interoperability of the system needs to be put across the level of health organizations in Indonesia.

## 5. Conclusions

In this study, we attempt to evaluate the implementation of the health information system regarding dengue in Indonesia. In particular, we underscore how incoherence in multi-level governance in a country as large as Indonesia can indeed become a hindrance in the implementation of the health information system in different levels of agency. In technical terms, interoperability and versatility of the system in place should become a priority. However, its success also depends largely on the adaptability of the human resources to a novel way of collecting data and monitoring a system. The study also offers an initial assessment of an integrated information system in the ever-increasingly complex world that involves wider issues such as big data, smart system, machine learning and industry 4.0. We believe that this is where further health policy analysis needs to delve into.

## Figures and Tables

**Figure 1 ijerph-17-01795-f001:**
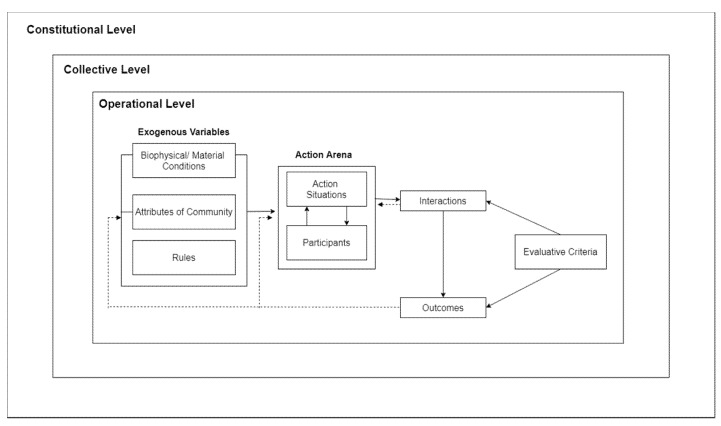
Institutional Analysis and Development (IAD) framework. The illustration is adapted from Ostrom (2005) [21].

**Figure 2 ijerph-17-01795-f002:**
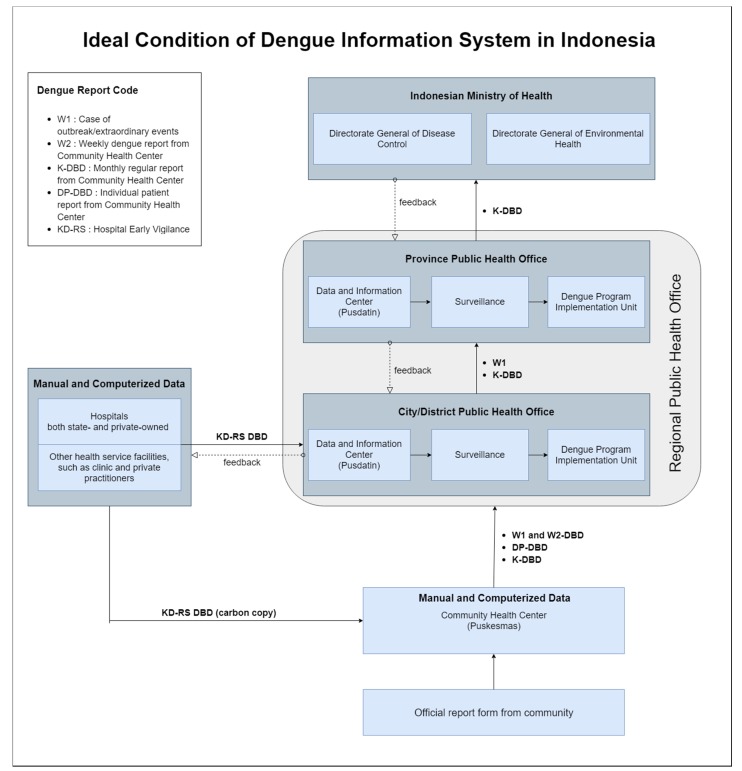
Ideal condition of the dengue health information system, as described by the Indonesian Ministry of Health [31].

**Figure 3 ijerph-17-01795-f003:**
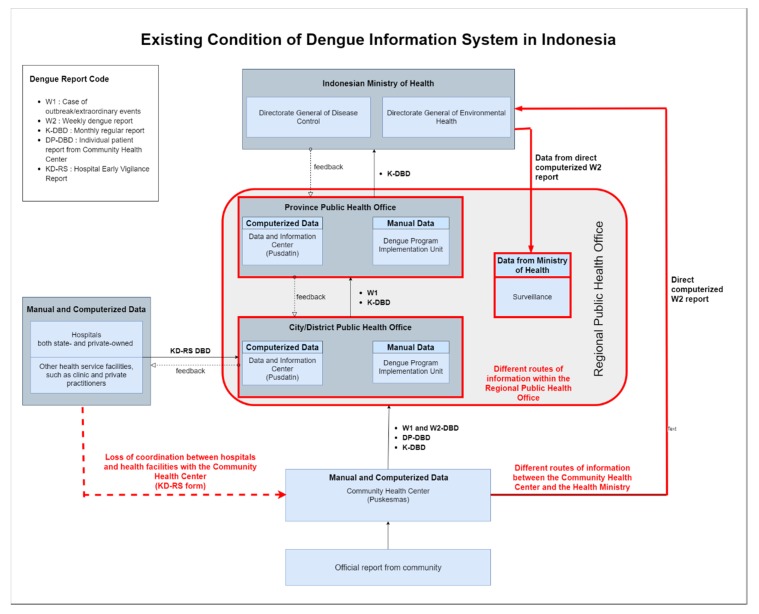
The existing condition found in Bandung, Indonesia regarding the implementation of the dengue health information system. Key differences between the ideal and the existing health information system are marked in red.

**Figure 4 ijerph-17-01795-f004:**
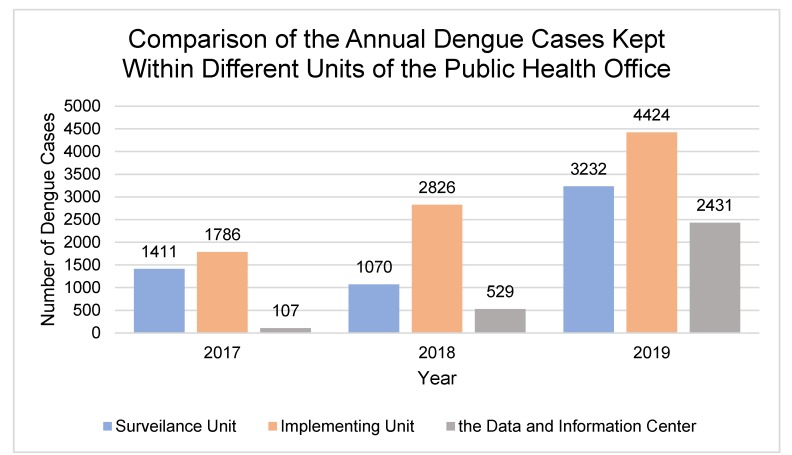
Comparison of annual dengue cases recorded by different units of the Public Health Office in 2017–2019. The implementing unit always records the highest number of cases while the Data and Information Center consistently records the least.

**Figure 5 ijerph-17-01795-f005:**
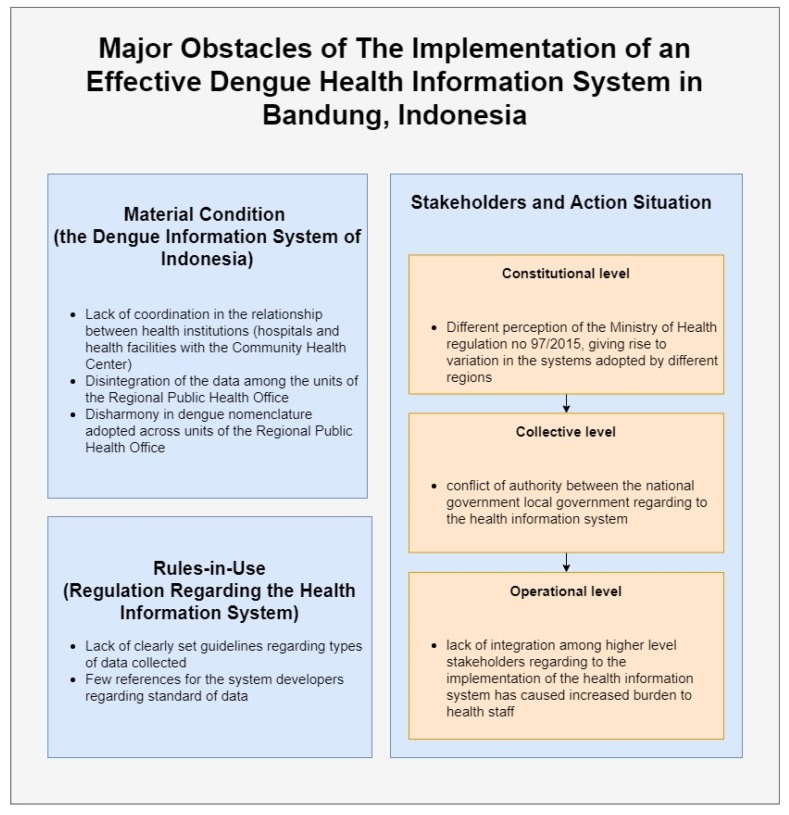
Summary of major obstacles identified that hinders an effective implementation of the health information system in dengue.

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
