# Peer review of "Evaluation of Health Information System (HIS) in The Surveillance of Dengue in Indonesia: Lessons from Case in Bandung, West Java"

_ijerph, 2020, doi:10.3390/ijerph17051795_

Round 1

Reviewer 1 Report

Specific comments have been made on the manuscript and authors should pay careful attention to implementing them. Of particular interest, and to improve the flow of the document, it will be good for the authors to pass the document through the hand of an English editor. 

The content is sound but the presentation sometimes made the reading and understanding complex.

Author Response

Thank you for your constructive review.

I have attached a point-by-point response, please see the attachment.

Reviewer 2 Report

This paper describes important issues in the control and management of endemic dengue fever in Indonesia. It has identified key challenges for authorities in a context of decentralisation of government responsibility and devolution of administrative and financial management as these relate to the operation of a Health Information System. 

There are two issues that need to be addressed. The first is in the methodological description. While broad categories of document analysis and key informant interview / focus group are identified, the approaches to the organisation and analysis of data acquired by these methods are not described.

Was thematic analysis applied to these texts? If so, what were the approaches to coding of document text and interview transcripts? Were these inductive (codes derived from the document/transcript texts) or deductive (codes derived from external sources such as monitoring frameworks or scholarly literature etc)? Or were both approaches used?

This should be detailed and specific reference to the themes attributed from the codes should be made when interpreting these themes (e.g. from your discussion: lack of coordination; incomplete data reporting; complexity of agency relationships etc etc) through the IAD framework. This is the key link between the data source and the justification of your recommendations and Conclusions.

The second issue is that there appears to be no mention of ethical review of the research. This review and approval reference should be mentioned in the Introduction

Author Response

(The authors gave the same response as above.)

Round 2

Reviewer 1 Report

The authors have addressed the issues of concern with regards to the manuscript. I have no further inputs. Well done.